# Quasi-weekly oscillation of regional PM$_{2.5}$ transport over China driven by the synoptic-scale disturbance of East Asian Winter Monsoon circulation

Yongqing Bai [1], Tianliang Zhao [2,*], Kai Meng [3,*], Yue Zhou [1], Jie Xiong [1], Xiaoyun Sun [4], Lijuan Shen [5], Yanyu Yue [1], Yan Zhu [1], Weiyang Hu [6], Jingyan Yao [2]

[1]China Meteorological Administration Basin Heavy Rainfall Key Laboratory/Hubei Key Laboratory for Heavy Rain Monitoring and Warning Research, Institute of Heavy Rain, China Meteorological Administration, Wuhan 430205, China

[2]Climate and Weather Disasters Collaborative Innovation Center, Key Laboratory for Aerosol-Cloud-Precipitation of China Meteorological Administration, Nanjing University of Information Science &Technology, Nanjing 210044, China

[3]Key Laboratory of Meteorology and Ecological Environment of Hebei Province, Hebei Provincial Institute of Meteorological Sciences, Shijiazhuang, 050021, China

[4]Anhui Province Key Laboratory of Atmospheric Science and Satellite Remote Sensing, Anhui Institute of Meteorological Sciences, Hefei 230031, China

[5]School of Atmosphere and Remote Sensing, Wuxi University, Wuxi, 214105, China

[6]State Key Laboratory of Pollution Control and Resource Reuse and School of the Environment, Nanjing University, Nanjing 210023, China

*Correspondence to*: Tianliang Zhao (tlzhao@nuist.edu.cn) and Kai Meng (macka@foxmail.com)

**Abstract:** The regional PM$_{2.5}$ transport is one of the important causes for atmospheric environment change. However, the variations of regional PM$_{2.5}$ transport in synoptic scale with meteorological drivers have been incomprehensively understood. Therefore, this study is targeted at the quasi-weekly oscillation (QWO) of regional PM$_{2.5}$ transport over central and eastern China (CEC) with the influence of synoptic-scale disturbance of the East Asian Winter Monsoon (EAWM) circulation. By constructing the data of daily PM$_{2.5}$ transport flux in CEC in the winters of 2015-2019, we utilize the extended empirical orthogonal function (EEOF) decomposition and other statistical methods to extract the moving spatial distribution of regional PM$_{2.5}$ transport over CEC, recognizing the QWO in regional PM$_{2.5}$ transport with the spatial-temporal variations over CEC. The source-acceptor relationship in regional transport of PM$_{2.5}$ is identified with the 2-d lag effect of the North China Plain, as the upwind source region, on the PM$_{2.5}$ pollution change in the Twain-Hu Basin, as the downwind receptor region in central China. The QWO of regional PM$_{2.5}$ transport over CEC is regulated by the synoptic-scale disturbance of the EAWM circulation with the periodic activities of Siberian high. These findings could provide new insight into the

understanding of regional PM$_{2.5}$ transport with source-receptor relationship and the meteorological

mechanism in atmospheric environment change.

**Key words:** regional PM$_{2.5}$ transport, quasi-weekly oscillation, source-receptor relationship, extended empirical orthogonal function (EEOF)

**1 Introduction**

PM$_{2.5}$ pollution has attracted worldwide attention due to its adverse impact on the environment and human health (Fan et al., 2016; Geng et al., 2021; Lin et al., 2018). The PM$_{2.5}$ pollution in the cold season has become one of the major atmospheric environmental problems in China (An et al, 2019; Huang et al, 2020b). The high-concentration PM$_{2.5}$ tends to occur with extensive spatiotemporal coverage (Tao et al, 2016; Zhang et al, 2019), and synthetic physical-chemical processes caused such heavy PM$_{2.5}$ pollution events (Ding et al, 2017; Quan et al., 2020), including emissions (Liu et al, 2016; Zheng et al, 2018a), chemical formation (Huang et al, 2014; Nie et al, 2014), atmospheric boundary layer processes (Huang et al, 2018; Zhong et al, 2019), localized circulation (Miao et al, 2015; Shu et al, 2021; Zheng et al, 2018b), as well as weather and climate (Cai et al, 2017; Wu et al, 2016). The interactions among these physical and chemical processes make it more challenging to comprehend the severe haze formation, which serves as one of the major difficulties in forecasting and controlling atmospheric environment change and heavy air pollution (Zhang et al., 2012; Zhang et al., 2019).

PM$_{2.5}$ is featured with complex spatiotemporal changes on multiscale (Georgoulias and Kourtidis, 2012; Wu et al, 2021). PM$_{2.5}$ oscillates periodically at multi-time scales, and the periodic oscillation of atmospheric circulation is the leading cause of the cyclical variations of PM$_{2.5}$ (Chen et al, 2020; Dong et al, 2021; Fu et al, 2020; Perrone et al, 2018). To be specific, the 1-d periodic change or diurnal variation of near-surface PM$_{2.5}$ concentrations is mainly attributed to the atmospheric boundary layer process and localized circulation (Miao et al, 2019); the periodic change of around 7 days may be controlled by the fluctuation of the long-wave trough in middle and high latitudes (Guo et al, 2014); the oscillating cycle of about 14 days is closely related to the quasi-biweekly oscillation of the synoptic circulation (Gao et al, 2020; Zhao et al, 2019); and the 30-60-d intra-seasonal oscillation is mainly caused by the impact of monsoon

circulation change (Xu et al, 2014; Zhang et al, 2019). Comprehensively revealing the interaction between $PM_{2.5}$ and meteorology at different time scales is essential for solving air pollution problems more effectively (Bäumer and Vogel, 2007; Wang et al, 2020). Previous studies mainly focused on the multiscale periodic variation of atmospheric pollutants in a certain region or local area, have not yet found on the $PM_{2.5}$ trans-regional and periodic oscillation in the large area of central and eastern China (CEC).

East Asian Winter Monsoon (EAWM) is one of the most active atmospheric circulation system in the cold season over the Northern Hemisphere (Ding et al, 2017; Wu and Wang, 2002), which is also a critical leading factor for the variation of wintertime air pollution in CEC (Chin, 2012; Li et al, 2016). Being the major circulation system of EAWM, the Siberian High dominates the cold seasons, acting as a particular driver of cold airflows, so having an important impact on the wintertime atmospheric environment in CEC (An et al, 2019; Shen et al, 2021, 2022; Wu et al, 2016). The rapid southward advance of cold air with strong Siberian High can effectively drive the regional transport of air pollutants with less accumulations across CEC, while the weak Siberian High with the slow southward movement of cold air can particularly favorable for the transport of air pollutants from the northern source regions to southern receptor region over CEC (Hou et al., 2020; Zhang et al., 2016). When the position of Siberian High is more eastern than normal, the transport of air pollutants from northern China to the south is weakened, and the aggravation of pollution is enhanced in northern China (Jia et al., 2015). Regional pollutant transport driven by the southward movement of a cold front with the Siberian High would exacerbate the air quality in the corresponding receptor regions (Kang et al., 2019; Hu et al., 2021; Shen et al, 2022). The characteristics of atmospheric circulation anomalies favoring heavy haze pollution in China have changed in recent years, and the leading formation mechanism of severe haze has been shifting from local accumulation to regional transport processes in eastern China (Yang et al, 2021b). Therefore, studying the influence of EAWM circulation system on regional pollutant transport over CEC is an important issue in atmospheric environment changes (Bai et al, 2021, 2022; Ge et al, 2018; Merrill and Kim, 2004; Tan et al, 2021; Yang, et al, 2021a).

Previous studies have primarily focused on the relationship between atmospheric intraseasonal oscillations in the mid-to-high latitudes of the Eurasian region and the persistent $PM_{2.5}$ pollution (An et al., 2022; Gao et al., 2020; Li et al., 2021; Liu et al., 2022; Wu et al., 2023;

Yang et al., 2024b). $PM_{2.5}$ concentration anomalies in North China exhibit significant lifetimes of
10–30 days, with anticyclonic anomalies and related meteorological conditions (e.g., surface air
temperature, boundary layer height) in Northeast Asia influencing local $PM_{2.5}$ accumulation and
hygroscopic growth (An et al., 2022; Yang et al., 2024b). These studies have investigated the
quasi-biweekly lifecycle of persistent $PM_{2.5}$ pollution events in North China through phase
synthesis methods (Gao et al., 2020; Wu et al., 2023; Yang et al., 2024b). However, there remains
a lack of systematic studies on the synoptic-scale oscillation of regional $PM_{2.5}$ transport.
The "harbor" effect on the eastern lee of the Tibetan Plateau's large topography on the
westerlies is possibly an important factor influencing the regional distribution of $PM_{2.5}$ pollution
in CEC with weak horizontal winds and sinking motion in the lower troposphere, which
exacerbates the environmental impacts of local air pollutant emissions establishing a
"susceptibility zone" in this region (Xu et al., 2016; Zhu et al, 2018). Anticyclones and cyclones
alternatively affect the region on a time scale of 3-7 days, resulting in periodic air pollution in
cities (Guo et al., 2014). Thus, the weather system in the CEC is basically characterized by
periodic changes and the cold air in winter with EAWM oscillates in quasi-weekly periods (Wu
and Wang, 2002; Wu et al., 2016). However, the influence of the synoptic-scale disturbance of the
EAWM on regional $PM_{2.5}$ transport over CEC is not yet clear. Responding to this problem, this
study aims to reveal from a new perspective the quasi-weekly oscillation (QWO) of regional $PM_{2.5}$
transport over CEC affected by EAWM and its underlying mechanism with the synoptic-scale
oscillation of the EAWM circulation. This study could deepen the understanding of regional $PM_{2.5}$
transport, its source-receptor relationship and meteorological mechanism in the atmospheric
environment changes, and provide scientific evidence for air pollution forecast, early warning and
coordinated control.

**2 Data and methods**
2.1 Environmental and meteorological data

The daily dataset of $PM_{2.5}$ concentrations selected for this study was from China National
Environmental Monitoring Center (http://datacenter.mee.gov.cn/), including daily $PM_{2.5}$
concentrations from 1079 air quality monitoring stations in CEC during the winters
(December-February) of 2015-2019.
Meteorological data were selected out of the NCEP/NCAR global reanalysis daily data
(https://psl.noaa.gov/data/gridded/tables/daily.html) with a grid resolution of 2.5 °×2.5 ° for the
large-scale circulation analysis. It is composed of the daily sea level pressure (SLP), air
temperature at 1000 hPa, and the U- and V-components of wind at 1000 hPa during the winters of
2015–2019.
In    addition,    the    ERA5-land    high-resolution    reanalysis    hourly    dataset
(https://cds.climate.copernicus.eu/cdsapp#!/dataset/reanalysis-era5-land?tab=form)    with    spatial
resolution of 0.1 °×0.1 ° was selected for the calculation of transport flux (TF) of $PM_{2.5}$ in CEC.
The U- and V-components of the 10-m wind over CEC were obtained at 00, 06, 12, and 18 UTC
daily during the winter (December-February) of 2015-2019. In order to match the resolution of
$PM_{2.5}$ daily data, the ERA5-Land high-resolution 10-m wind was processed into daily average
data.

2.2 $PM_{2.5}$ TF and its divergence

In order to quantitatively characterize the horizontal transport direction and intensity of $PM_{2.5}$
as well as convergence or divergence during regional $PM_{2.5}$ transport, we introduced the concepts
of $PM_{2.5}$ TF and divergence of $PM_{2.5}$ TF. Generally, there are two types of TF: horizontal and
vertical. This study only addresses the near-surface horizontal $PM_{2.5}$ TF. The horizontal $PM_{2.5}$ TF
is defined as the $PM_{2.5}$ mass passing through the unit area in unit time (unit: $\mu g\ m^{-2}\ s^{-1}$), expressed
as the product of wind vector and $PM_{2.5}$ concentration (Liu et al., 2019; Ma et al., 2021), and its
vector points to the same direction as the horizontal wind. The zonal component ($F_u$) and
meridional component ($F_v$) of $PM_{2.5}$ TF vector (TFV) and the magnitude (TFM) are calculated as
follows:
$$F_u = C\ u \tag{1}$$

$$F_v = C\ v \tag{2}$$

$$TFV = F_u\ i + F_v\ j \tag{3}$$

$$TFM = \sqrt{F_u^2 + F_v^2} \tag{4}$$

where $C$ is the surface $PM_{2.5}$ concentration, $u$ and $v$ are the zonal and meridional components
of the 10-m wind speed, respectively.

Firstly, the U- and V-components of ERA5-Land high-resolution 10-m wind are interpolated

to 1079 stations of environmental measurements in CEC for calculations of near-surface $PM_{2.5}$ TF
in this study. Then, the daily $PM_{2.5}$ TF of the 1079 stations for the winters from 2015 to 2019 are
calculated according to the calculation by Formulas (1)–(4).

The divergence of $PM_{2.5}$ TF can be an indicator for the $PM_{2.5}$ budget. When positive

divergence occurs, the air pollutants were net outflow from the domain region, and vice versa
(Wang et al., 2021). The divergence of horizontal $PM_{2.5}$ TF near the surface is calculated as
follows (Wang et al., 2021):
$$D = \frac{\partial F_u}{\partial x} + \frac{\partial F_v}{\partial y} \qquad (5)$$
where $D$ is the horizontal $PM_{2.5}$ TF divergence, unit: $\mu g\,m^{-3}\,s^{-1}$. If $D$ is positive (negative), it
indicates divergence (convergence) of $PM_{2.5}$ TF.

In the $i$ and $j$ grids, the expression of Formula (5) for the differential calculation with grid

spacing to be $d$ is
$$D = \frac{Fu_{i+1,j} - Fu_{i-1,j} + Fv_{i,j+1} - Fv_{i,j-1}}{2d} \qquad (6)$$

When calculating the horizontal divergence of transport $PM_{2.5}$ flux, it is necessary to

interpolate the station data of zonal and meridional components ($F_u$, $F_v$) of $PM_{2.5}$ TFV to grid
spacing with 0.25 by 0.25 degree in longitude and latitude in CEC and then calculate the
divergence of $PM_{2.5}$ TF at each grid point according to Formula (6).

2.3 Butterworth filter

Atmospheric motion encompasses a variety of temporal and spatial scales. The sequences of

meteorological variables often contain complex periodic components and exhibit multi-time-scale
variations, including daily, weekly, seasonal, and interannual variations. Numerous observations
have found QWO with periods of less than 10 days across various meteorological elements in the
EAWM system (Compo et al., 1999; Murakami, 1979; Wu and Wang, 2002). Synoptic-scale
atmospheric variations are closely related to atmospheric longwave adjustments, with QWO
periods of 4-7 days observed in cold air activities of the EAWM (Bai et al., 2022; Wu and Wang,
2002). The synoptic-scale disturbance regulates the generation, transport, and removal of $PM_{2.5}$ in
air pollution, which is a key mechanism behind the 4-7 day periodic changes in $PM_{2.5}$ in CEC
during the periods of EAWM (Guo et al., 2014; Liu et al., 2018; Quan et al., 2014, 2020). Based
on the research objectives, identifying the desired periodic components from the original
observational sequences is referred to as sequence filtering. In this study, we employed a
Butterworth filter to extract QWO from observational data.
The Butterworth filter is commonly used to separate atmospheric periodic variations across
specific frequency bands. Due to its smooth amplitude response, linear phase characteristics, and
ease of implementation, Butterworth filter has been widely applied in climate and meteorological
studies (Gouirand et al., 2012; Yang et al., 2024a). The Butterworth filter can be configured as a
low-pass, high-pass, or band-pass filter, depending on the specific requirements. A band-pass
filtering only allows signals within a defined frequency range to pass through with attenuating
signals outside the defined frequency range. It is often employed to extract and analyze signals
within specific frequency bands, such as particular weather patterns and climate cycles. In this
study, to investigate the QWO (8-d) of regional $PM_{2.5}$ transport over the CEC under the influence
of EAWM circulations in the synoptic scale, we applied Butterworth band-pass filtering to the
daily TFM of $PM_{2.5}$ change and daily SLP anomalies during the winters of 2015-2019 for
identifying at the quasi-weekly (6-9 days) synoptic-scale component of regional transport of
$PM_{2.5}$ over CEC.

2.4 Extended empirical orthogonal function (EEOF)

The Empirical Orthogonal Function (EOF) analysis is a widely-applied climate statistical
method in atmospheric and oceanographic scientific studies (Kim et al., 2015; Li et al., 2019;
Schepanski et al., 2016), also used to investigate the variability of atmospheric aerosols at
different spatiotemporal scales (Bai et al., 2022; Feng et al., 2020). The mathematical process of
EOF analysis is to decompose the variable field $X_{m \times n}$, which consists of observations at $n$ times at
$m$ spatial points, into a linear combination of $p$ spatial eigenvectors (modes) with corresponding
time-weighting coefficients:
$$X_{m \times n} = V_{m \times p} \; T_{p \times n} \tag{7}$$

where $V$ is the spatial eigenvector (load) and $T$ represents the time coefficient. The main
information of variable field $X_{m \times n}$ is represented by several eigenvectors. Since the method has
been maturely applied, the detailed calculation steps of EOF decomposition are omitted here, and
our focus is on how to construct the observation matrix.
Firstly, we decompose the daily $PM_{2.5}$ TFM anomalies of 1079 stations in CEC during the
winters of 2015-2019 by EOF method. Thus, the following observation matrix can be obtained:

$$
\qquad X = \begin{bmatrix} X_{11} & \cdots & X_{1n} \\ \vdots & & \\ X_{m1} & \cdots & X_{mn} \end{bmatrix}
$$

(8)

where $X$ represents the $PM_{2.5}$ TFM anomalies, $m$ represents the spatial points for 1079
stations, and $n$ represents the observation times of 450 days. Then, the variable field $X$ is
decomposed into the sum of the product of space and time functions according to Formula (7).
EOF decomposition of $PM_{2.5}$ TFV anomalies can be performed by employing the complex
matrix, hence the following observation matrix is constructed:

$$
\qquad X = \begin{bmatrix} u_{11} & \cdots & u_{1n} \\ \vdots & & \\ u_{m1} & \cdots & u_{mn} \\ v_{11} & \cdots & v_{1n} \\ \vdots & & \\ v_{m1} & \cdots & v_{mn} \end{bmatrix}
$$

(9)

where $X$ is the $PM_{2.5}$ TFV anomalies, and $u$ and $v$ refer to the zonal and meridional
components of TFV anomalies.
With EOF analysis we can get the spatial distribution structure, which is in a fixed time
pattern of climate variables, but we cannot get a temporally moving spatial distribution structure.
EEOF is an extension of the EOF to analyze the autocorrelations of the variable field over time.
By selecting a lag time, the original observational matrix is expanded into multiple continuous
time matrices, diagnosing the temporal changes in the spatial structure of variable fields. This
method has widespread applications in the analysis and prediction of marine and atmospheric
motions (Dey et al., 2018; Qian et al., 2019; Wang et al., 2019).
In this study, we utilized the EEOF analysis to reveal the evolution of $PM_{2.5}$ TF to reveal the
spatiotemporal variations of regional $PM_{2.5}$ transport. On the basis of Formula (8), a new extension
matrix of $PM_{2.5}$ TFM is constructed. Due to the study on the synoptic scale, 5 lag times are
selected, and each lag time is 1 day in length. The constructed observation matrix is as follows:

$$X = \begin{bmatrix} X_{1,1} & \cdots & X_{1,n-5} \\ \vdots & & \\ X_{m,1} & \cdots & X_{m,n-5} \\ X_{1,2} & \cdots & X_{1,n-4} \\ \vdots & & \\ X_{m,2} & \cdots & X_{m,n-4} \\ X_{1,3} & \cdots & X_{1,n-3} \\ \vdots & & \\ X_{m,3} & \cdots & X_{m,n-3} \\ X_{1,4} & \cdots & X_{1,n-2} \\ \vdots & & \\ X_{m,4} & \cdots & X_{m,n-2} \\ X_{1,5} & \cdots & X_{1,n-1} \\ \vdots & & \\ X_{m,5} & \cdots & X_{m,n-1} \\ X_{1,6} & \cdots & X_{1,n} \\ \vdots & & \\ X_{m,6} & \cdots & X_{m,n} \end{bmatrix} \quad (10)$$

Seen from Formula (10), the new extended matrix is composed of $X_{6m,n-5}$, where $X$ is the
PM$_{2.5}$ TFM anomalies, $m$ is the spatial points of observation station, and $n$ is the observation times
of 450 days. When EEOF decomposition is performed on PM$_{2.5}$ TFV, the complex matrix is still
used for the extension, and the same lag scheme is adopted to construct a new extended matrix of
PM$_{2.5}$ TFV based on Formula (9). After constructing the initial data matrix, the EEOF
decomposition method is in line with the classical EOF decomposition method.
Additionally, existing studies have utilized wavelet analysis, power spectrum analysis, and
band-pass filtering methods to extract intraseasonal oscillation sequences of regional PM$_{2.5}$
concentrations (An et al., 2022; Gao et al., 2020; Li et al., 2021; Liu et al., 2022; Wu et al., 2023;
Yang et al., 2024b). Such approaches may serve as alternative methods to EEOF analysis for
establishing the quasi-weekly lifecycle of regional PM$_{2.5}$ transport.

**3 Results and discussion**

3.1 QWO of regional PM$_{2.5}$ transport over CEC

The EOF decomposition is carried out on the daily anomalies of PM$_{2.5}$ TFM and TFV in the

winters of 2015-2019 over CEC. The first two EOFs explain 26.6% and 14.2% (29.1% and 11.8%)
of the total anomalous variations of PM$_{2.5}$ TFM (TFV), which is very helpful for better
characterizing regional PM$_{2.5}$ transport variations.

Two principal modes govern the variations of PM$_{2.5}$ TF anomalies over CEC: the first leading

mode of monopole (EOF1) and the second mode of meridional dipole (EOF2) (Fig. 1). EOF1
indicates the enhanced PM$_{2.5}$ TF over CEC (Fig. 1a). The large value center of TF mainly occurs
in central China, and the transport vector direction is abnormally by north. The horizontal PM$_{2.5}$
transport is unusually strong in central China affected by the EAWM, presenting a typical channel
for regional PM$_{2.5}$ transport over CEC (Yang et al., 2021a). The dipole mode of PM$_{2.5}$ TF
anomalies displays a south–north out-of-phase pattern, with the flux large value centers located in
the North China Plain (NCP) and the Twain-Hu Basin (THB) respectively, and the vector
directions are opposite (Fig. 1b). This mode indicates that the air pollutants from NCP in the
upwind are transported to THB in the downwind driven by the prevailing northerlies of EAWM
(Hu et al., 2021; Shen et al., 2022), and the PM$_{2.5}$ flux in NCP decreases while that in THB
increases in the regional PM$_{2.5}$ transport process.

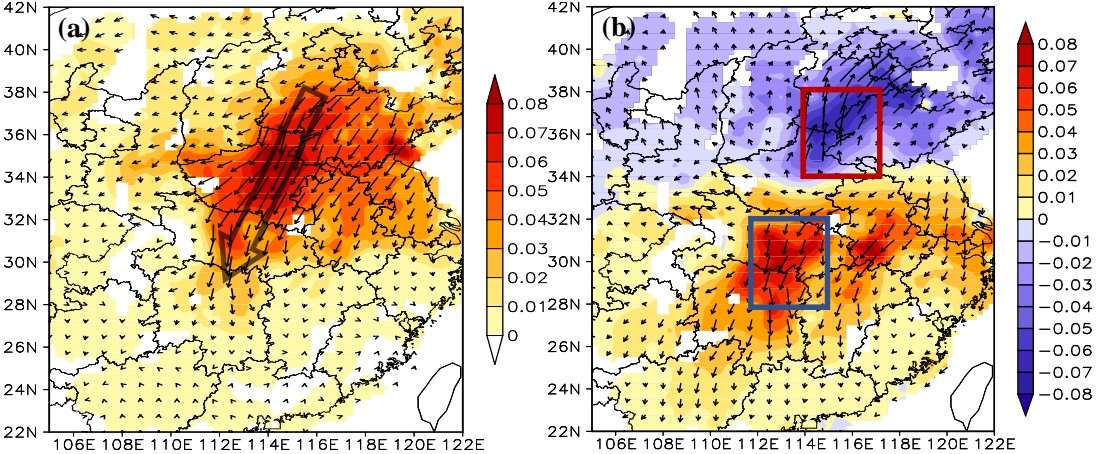


**Figure 1.** Spatial pattern of the (a) EOF1 and (b) EOF2 loads in the daily change of PM$_{2.5}$ TFV anomalies (vectors,
unitless) and TFM anomalies (color contours, unitless) over CEC in the winters of 2015-2019. The red and blue
boxes indicate NCP and THB, respectively. The grid cells in white represent "missing values".

Through EOF decomposition, the $PM_{2.5}$ TF could be understood from the perspective of a
fixed time pattern of climate, but the temporal changes in the moving spatial structure of $PM_{2.5}$ TF
over CEC failed to be obtained. However, EEOF decomposition can be used to analyze the
continuous structural evolution of the main modes of regional $PM_{2.5}$ TF over CEC.
The EEOF decomposition was carried out for the daily variations of $PM_{2.5}$ TFM anomalies
and TFV anomalies respectively over CEC during the winters of 2015-2019. Figure 2 and Figure
S1 show the spatial distribution of different lag times for the main modes of EEOFs, which
account for about 20% of the total variation. According to the analysis, the $PM_{2.5}$ TFM anomalies
for EEOF2 and EEOF3, as well as TFV anomalies for EEOF1 and EEOF2, all show the structural
evolutions in the different phases of regional $PM_{2.5}$ transport in one cycle. As it can be seen,
Figures. 2a-d, S1a-d, and 2e-h respectively describe the evolution of the first and second four
phases in a cycle and the first four phases in the next cycle (one phase represents 1day).
Figures 2a-d illustrate the positive anomalies of $PM_{2.5}$ TF shifting from NCP to THB in the
first four phases under the effect of the EAWM, causing the upwind $PM_{2.5}$ TF to decrease and the
downwind $PM_{2.5}$ TF to increase, which is in line with the spatial pattern of the EOF modes in
Figure 1. The last four phases show the out-of-phase pattern of the first half cycle (Figs. S1a-d). It
is noted that when anomalies of $PM_{2.5}$ TFV in the NCP turn to the northerly direction (Fig. S1d
and Fig. 2e), it is a strong signal initiating the regional $PM_{2.5}$ transport. Then, the transport is
repeated in the next periodic cycle (Figs. 2e-h). Therefore, the regional $PM_{2.5}$ transport over CEC
enjoys a quasi-weekly (8-d) oscillation pattern.

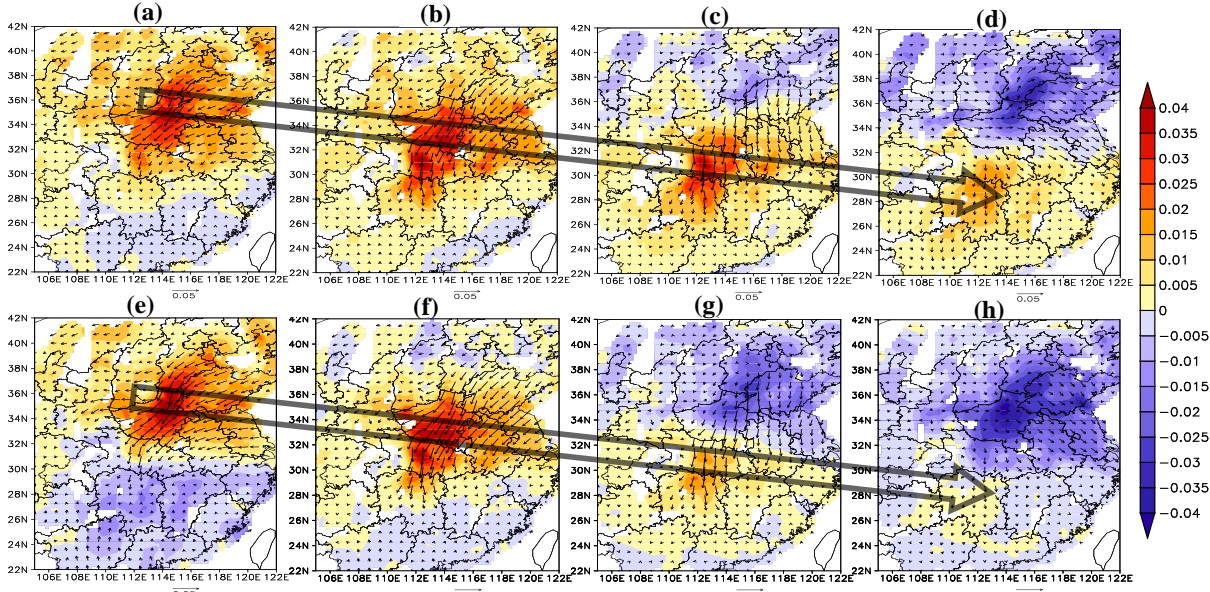

**Figure 2.** (a)-(d) The first four phases (days) of QWO (8-d) during the regional $PM_{2.5}$ transport over CEC; (e)-(h) the first four phases (days) of the next cycle. The Loads of $PM_{2.5}$ TFM anomalies (color contours, unitless) for EEOF2 and TFV anomalies (vectors, unitless) for EEOF1 with lag time (a) 0 d, (b)1 d, (c) 2 d and (d) 3 d, and loads of TFM anomalies (color contours, unitless) for EEOF3 and TFV anomalies (vectors, unitless) for EEOF2 with lag time (e) 2 d, (f) 3 d, (j) 4 d and (h) 5 d over CEC in the winters of 2015-2019.

To further study the variations of regional $PM_{2.5}$ transport over CEC, we have screened out 23 typical events with greater than 1.5 times standard deviations based on the standardized time coefficient of EEOF, and then used the 8 consecutive days of each event as the 8 phases of QWO in the composite analysis on the 23 typical events of regional $PM_{2.5}$ transport over CEC.

Figure 3 shows the composited $PM_{2.5}$ TF, divergence of $PM_{2.5}$ TF, and $PM_{2.5}$ concentration anomalies in the first four phases of QWO. The high fluxes of $PM_{2.5}$ transport from north to south persists for 3-4 days over CEC and decline in the THB (Fig. 3a-d). The regional $PM_{2.5}$ transport lifetime corresponding to synoptic systems is about 3-5 days (Huang et al., 2020a). Abnormal northerly winds drive the heavy $PM_{2.5}$ pollution from the upwind NCP to the downwind regions, aggravating $PM_{2.5}$ pollution in the downwind THB (Figs. 3e-h). Under the context of QWO, the average $PM_{2.5}$ TFM in NCP decreases from approximately 400 μg m$^{-2}$ s$^{-1}$ in the 1st and 2nd phases to 200 and 100 μg m$^{-2}$ s$^{-1}$ in the 3rd and 4th phases, respectively (Fig. S2a). Correspondingly, the $PM_{2.5}$ concentration anomalies decline from around 100 μg m$^{-3}$ to approximately −50 μg m$^{-3}$ (Fig. S2c). In the downwind THB, the average $PM_{2.5}$ TFM increases from about 200 μg m$^{-2}$ s$^{-1}$ in the

1st phase to approximately 300 μg m$^{-2}$ s$^{-1}$ in the 2nd and 3rd phases (Fig. S2b), with PM$_{2.5}$
concentration anomalies also rising to around 50 μg m$^{-3}$ (Fig. S2d).
It is noteworthy that the regions PM$_{2.5}$ TF convergence zone (negative value of divergence)
matches spatially the centers positive anomaly centers of PM$_{2.5}$ concentrations, which is confirmed
with a significantly negative correlation of the PM$_{2.5}$ concentrations with divergences of PM$_{2.5}$ TF
in the 23 typical events (Fig. S3). The PM$_{2.5}$ transport is accompanied by flux convergence, which
is beneficial to the PM$_{2.5}$ accumulation. In addition, the PM$_{2.5}$ TF in the upwind NCP changes
from convergence to divergence, and the divergence of the PM$_{2.5}$ TF in the downwind THB alters
to convergence in the meantime (Figs. 3i-l), indicating that the PM$_{2.5}$ over THB is transported
from the upwind NCP.

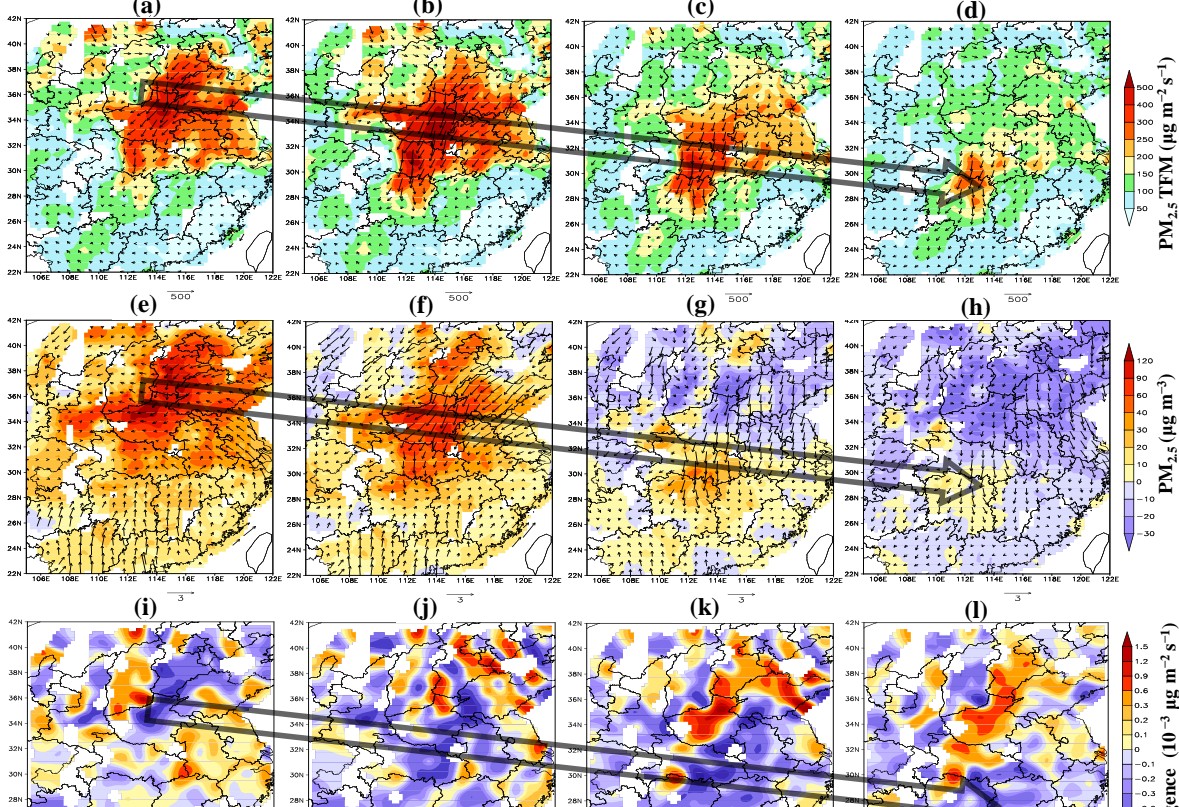

**Figure 3.** Spatial distributions of the composited (a-d) PM$_{2.5}$ TFM (color contours, unit: μg m$^{-2}$ s$^{-1}$) and TFV
(vectors, unit: μg m$^{-2}$ s$^{-1}$), (e-h) anomalies of PM$_{2.5}$ concentrations (color contours, unit: μg m$^{-3}$) and 10-m wind
vectors (unit: m s$^{-1}$), (i-l) divergence of PM$_{2.5}$ flux (color contours, unit: 10$^{-3}$ μg m$^{-3}$ s$^{-1}$) in the first four phases of
QWO during the 23 typical events of regional PM$_{2.5}$ transport over CEC.

3.2 Source-receptor relationship in regional PM$_{2.5}$ transport from NCP to THB

The regional pollutant transport governed by emissions and meteorology leads to a complex
source–receptor relationship of air pollution changes (Yu et al., 2020). Band-pass filtering is
performed on the daily PM$_{2.5}$ TFM anomalies at a quasi-weekly (6-9 days) synoptic scale in the
winters of 2015-2019. In Figure 4a, we composite the filter components of PM$_{2.5}$ TFM in the 8
phases of QWO during the 23 typical events of regional PM$_{2.5}$ transport over the NCP and THB,
respectively. The PM$_{2.5}$ TF exhibits an obvious QWO on the synoptic scale (Fig. 4a). The PM$_{2.5}$
TF over the NCP continues to decline in the first four phases, while that of THB first rises and
then falls in the last four phases, the PM$_{2.5}$ TF over the NCP increases continuously, while that of
THB falls first and then rises. We can see that the QWO of PM$_{2.5}$ TF over THB lags behind the
NCP by 2 phases (Fig. 4a). The high TFM of PM$_{2.5}$ from NCP in the first phase spread to THB,
resulting in the peak of PM$_{2.5}$ TF over THB in the third phase.
In addition, the distribution of the differences in PM$_{2.5}$ TF and the vectors between phase 3
and phase 1 of the QWO, and the PM$_{2.5}$ TF decrease and increase from phase 1 to phase 3
respectively over the upwind NCP and the downwind THB, which is in accordance with the
spatial pattern of the EOF mode (Figs.1b and 4b), indicating that the source-receptor relationship
over CEC exist the regions NCP and THB of regional PM$_{2.5}$ transport over CEC.

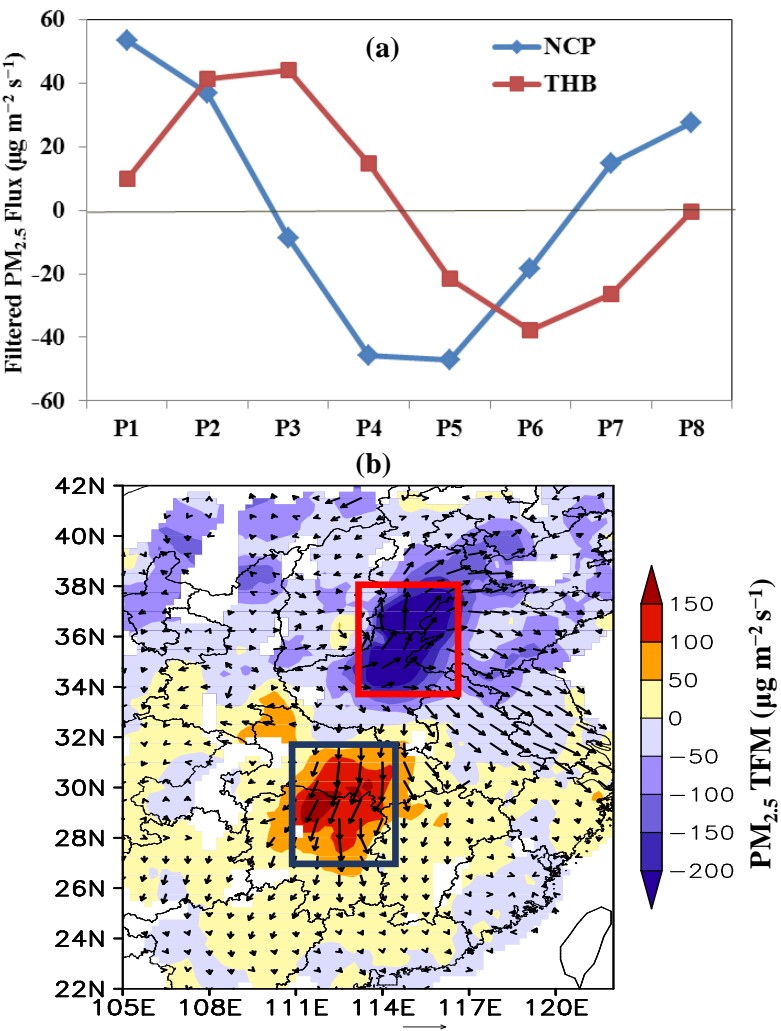


**Figure 4.** (a) The 8 phases (P1-P8) of QWO during the 23 typical events of regional PM$_{2.5}$ transport over the NCP

and THB with composited 6-9 d band-pass filtering of PM$_{2.5}$ TFM; (b) spatial distribution of the differences in

PM$_{2.5}$ TFM (color contours, unit: μg m$^{-2}$ s$^{-1}$) and TFV (vectors, unit: μg m$^{-2}$ s$^{-1}$) between the 3rd phase and the 1st

phase of QWO. The red and black boxes represent NCP and THB.

The statistical analysis based on long-term observation also shows that there is a significant

2-day lag relationship of positive correlation between NCP and THB in PM$_{2.5}$ TF in the QWO (Fig.

5a). This discloses that the air pollutants are transported from the upwind NCP to the downwind

THB in 2 days, confirming a quasi-2-d lag in the regional PM$_{2.5}$ transport from NCP to THB (Hu

et al., 2021; Shen et al., 2021). Additionally, in the long-term change of air pollution, the

divergences of PM$_{2.5}$ TF in the NCP are significantly negatively correlated to that of THB (Fig.

5b), that is, the PM$_{2.5}$ TF convergences in the downwind THB fits well with the PM$_{2.5}$ TF

divergence in the upwind NCP. It can be reflected that the changes in the synoptic scale of EAWM
atmospheric circulation impel the regional $PM_{2.5}$ transport to build the source-receptor relationship
of atmospheric pollutants between the NCP and THB.

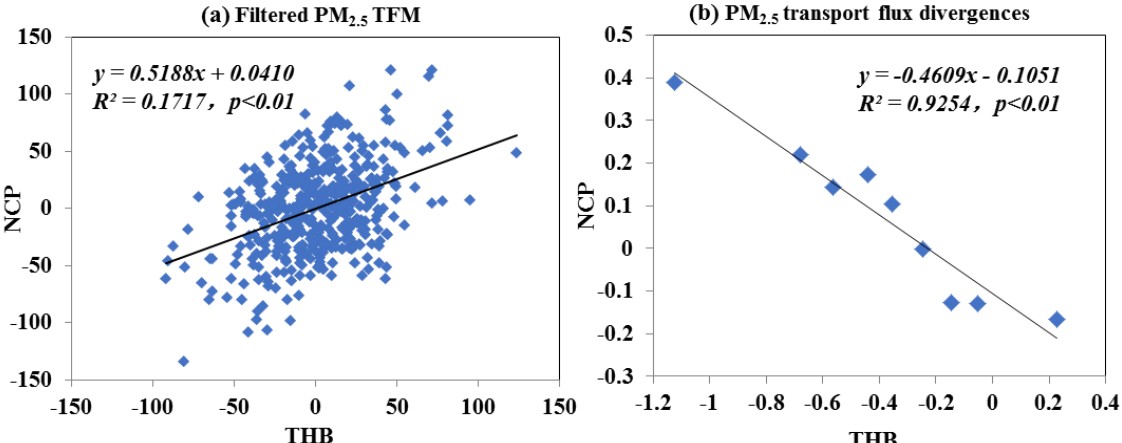

**Figure 5.** (a) Scatter plot of 6-9-d filtering components of $PM_{2.5}$ TFM ($10^{-3}$ μg m$^{-2}$ s$^{-1}$) over THB in 2-day lag and
NCP during the winters of 2015-2019; (b) scatter plot of $PM_{2.5}$ TF divergences ($10^{-3}$ μg m$^{-3}$ s$^{-1}$) between THB and
NCP, and the $PM_{2.5}$ TF divergences are averaged over the value interval of 0.1.

Driven by prevailing winds of EAWM, the THB became the main receptor for regional
transport of air pollutants over CEC (Bai et al., 2022; Shen et al., 2021). During 2015–2019,
approximately 65.2% of the total $PM_{2.5}$ heavy pollution events in the THB were triggered by
regional transport of air pollutants over CEC (Hu et al., 2022; Shen et al., 2021). Such $PM_{2.5}$
transport from upstream source regions in CEC contributes 51%-85.7% of the $PM_{2.5}$ pollution
over the THB receptor region (Hu et al., 2021; Lu et al., 2017; Shen et al., 2022; Yu et al., 2020),
revealing the dominance of regional transport of air pollutants from CEC to the THB with the
meteorological drivers. Our research emphasizes the QWO of regional $PM_{2.5}$ transport over CEC
with the driver of the synoptic-scale disturbances of EAWM circulation, confirming the
source-receptor relationships with their 2-day lagging effects in the regional $PM_{2.5}$ transport
between the upstream NCP source region and the THB receptor region.

3.3 Effect of synoptic-scale disturbance of EAWM circulation on QWO of regional $PM_{2.5}$ transport
over CEC

Meteorological change is the essential factor in regulating the occurrence and development of
PM$_{2.5}$ pollution on synoptic scales. To investigate the QWO of EAWM circulation in the synoptic
scale disturbance, this study performs the 6-9-d band-pass filtering of the daily SLP anomalies
(denoted as SLP$_{QWO}$) in East Asia during the winters of 2015-2019. The SLP and SLP$_{QWO}$ fields
(Figs. 6 and 7) as well as PM$_{2.5}$ concentrations and 10-m winds (Fig. S4) in the 8 phases of QWO
during the 23 typical events were composited, respectively. The QWO of regional PM$_{2.5}$ transport
is connected with the "weekly-cycle" synoptic process of PM$_{2.5}$ transport and accumulation over
CEC (Fig. S4), and it is powered mainly by the Siberian High circulation with the synoptic-scale
disturbance of EAWM circulation (Figs. 6 and 7).

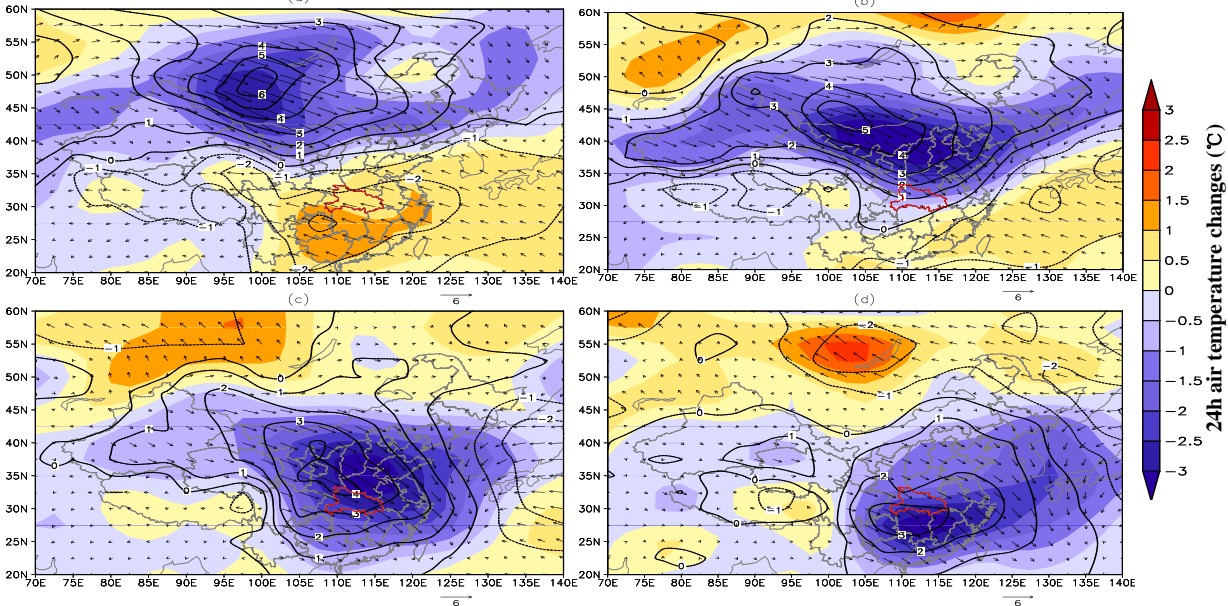


**Figure 6.** The composited differences between the current day and the previous day of SLP (black contour lines,

unit: hPa), 1000 hPa air temperature (color contours, unit: ℃) and wind vectors ( unit: m s$^{-1}$ ) in the first four

phases (a-d) of QWO during the 23 typical events.


The condition of uniform pressure in the front of Siberian High could favor the PM$_{2.5}$
accumulation over the NCP for triggering regional PM$_{2.5}$ transport over CEC (Fig. 7a). The
regional heavy pollution of PM$_{2.5}$ >150 μg m$^{-3}$ lasts for 1-2 days (Figs. S4a and S4b). With the
development of the Siberian High, the extension of the high pressure guides the cold air to
advance southward (Park et al., 2014). As the result of the increasing air pressure gradients, the
strong northerly winds in the EAWM circulation system, deliver high-level PM$_{2.5}$ air mass from
NCP to THB (Figs. 7a-d, Figs. S4a-d). In addition, the cold and high air pressure system with the
abnormal northerly airflows moves from the Siberia-Mongolia region to CEC in the first four
phases (Fig. 6), providing beneficial synoptic circulation patterns for regional $PM_{2.5}$ transport.
Thus, the periodic extension of the Siberian High with the associated strong cold air intrusion is an
important driver in the regional $PM_{2.5}$ transport over CEC.

Notably, we can see that in the first four phases, the $SLP_{QWO}$ positive anomalies occur,

develop, and expand southward from the Siberia-Mongolia region to CEC (Figs. 7a-d). The
synoptic-scale disturbance with the extension of Siberian High and the southward movement of
cold air could drive the regional $PM_{2.5}$ transport over CEC (Figs. 7a-d). The situation of the last
four phases is opposite to the $SLP_{QWO}$ negative anomalies in Siberia-Mongolia region, inhibiting
the Siberian High and cold air intrusion (Figs. 7e-h). The low and uniform pressure is beneficial to
the accumulation of $PM_{2.5}$. Therefore, the periodic changes in the synoptic-scale disturbance of the
EAWM circulation impel the QWO of regional $PM_{2.5}$ transport over CEC.


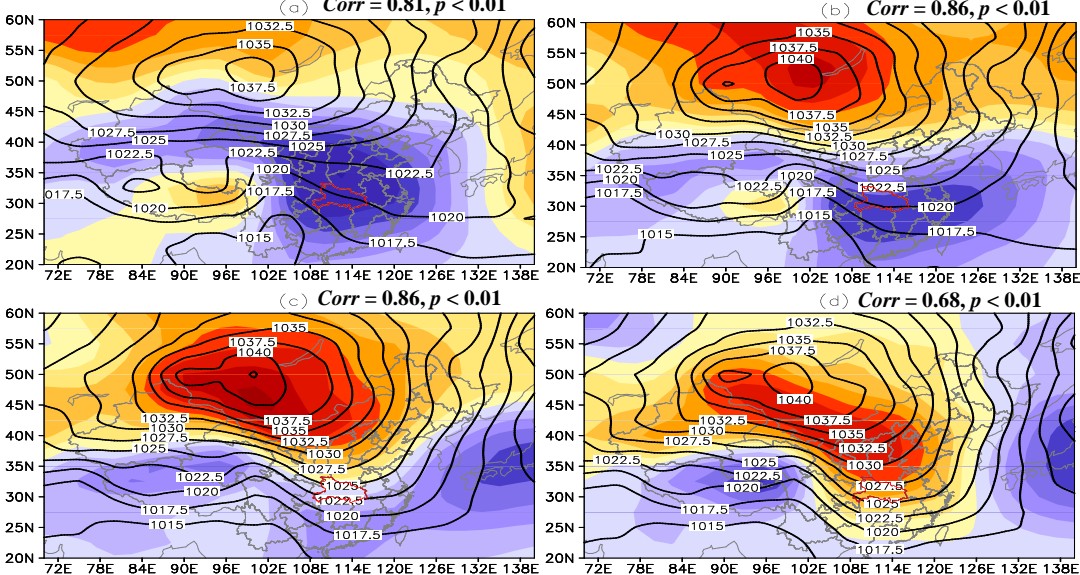

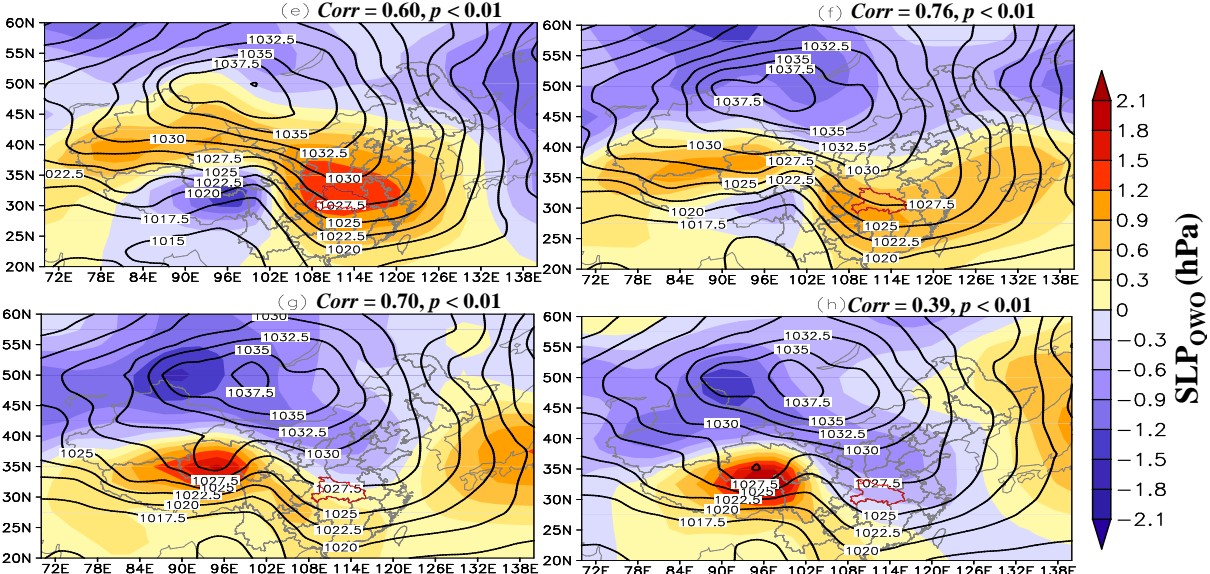

**Figure 7.** Composited SLP (black contour lines, unit: hPa) and its synoptic-scale filter component SLP$_{QWO}$ (color contours, unit: hPa) in the 8 phases (a-h) of QWO during the 23 typical events. *Coor* represents the spatial correlation coefficients between SLP$_{QWO}$ and the load of SLP$_{QWO}$ decomposed by EEOF in Fig. S4.

In addition, the EEOF decomposition is carried out on the SLP$_{QWO}$ field in the winters of 2015-2019 to recognize the periodic activities in the synoptic scale of the EAWM circulation. The cold air activity of EAWM presents QWO (Wu and Wang, 2002). The positive (negative) synoptic-scale disturbance occurs in the Siberia-Mongolia region, and then spreads to CEC along the northwest-southeast path, contributing to the 8-d cycle of QWO (Fig. S5). Notably, the spatial correlation coefficients between the load of SLP$_{QWO}$ decomposed by EEOF (Fig. S5) and the SLP$_{QWO}$ composited during 23 typical events (Fig.7) are highly positively correlated in the 8 phases, respectively. Therefore, the QWO in the synoptic-scale activities of the Siberian high is an important factor for driving the QWO of regional PM$_{2.5}$ transport over CEC.

## 4 Conclusions

Exploring the periodical oscillations of PM$_{2.5}$ pollution over CEC and the meteorological effect is crucial for understanding the change in the atmospheric environment and improving regional air quality forecasts. In this study with constructing a dataset of the daily PM$_{2.5}$ TF, the EEOF and statistical methods are used to identify the QWO of regional PM$_{2.5}$ transport with the spatiotemporal variations over CEC in winters from 2015 to 2019. The source-receptor

relationship is recognized between NCP and THB with the QWO of regional $PM_{2.5}$ transport over
CEC with the typical EAWM climate. Furthermore, it is revealed that the driving effect of
synoptic-scale disturbance of EAWM circulations on the QWO of regional $PM_{2.5}$ transport over
China.
The variations of $PM_{2.5}$ TF over CEC are dominated by the first leading monopole mode and
the second meridional dipole mode. The monopole mode indicates the high $PM_{2.5}$ flux along the
channel of regional $PM_{2.5}$ transport from NCP to THB under the governs of the EAWM
circulations, and the dipole mode exhibits a pattern of south-north out-phase with two centers
existing respectively in the upwind NCP and the downwind THB in regional transport of $PM_{2.5}$
over CEC. In terms of the long-term changes in air pollution of 2015–2019, the regional $PM_{2.5}$
transport over CEC is featured with the QWO, verifying a source-receptor relationship for the
regional $PM_{2.5}$ transport from NCP to THB in 2 days. Such changes are incurred by the QWO in
the activities of the Siberian High, and this synoptic-scale disturbance of the EAWM circulations
is generated in the Siberia-Mongolia region, and then develops, marching into CEC, regulating the
QWO of regional $PM_{2.5}$ transport.
The EEOF analysis with the temporal lag of the spatial fields is able to better characterize the
spatial and temporal evolution of perturbations, especially propagating waves in the atmosphere
(Weare and Nasstrom, 1982; Qian et al., 2019; Yang et al., 2024b). Due to its technical advantages,
the EEOF method is commonly employed to extract atmospheric oscillation patterns to reveal the
impacts and mechanisms of atmospheric fluctuations and monsoon circulation on regional weather,
climate, and atmospheric environments (Dey et al., 2018; Qian et al., 2019; Yang et al., 2024b). In
this study, we employed the EEOF method to identify regional $PM_{2.5}$ transport modes in synoptic
scale, by constructing $PM_{2.5}$ transport flux vectors (TFV) and the magnitude (TFM) with the
product of near-surface $PM_{2.5}$ concentrations and wind components at 1079 stations across China
during the winters of 2015-2019. We performed EEOF analysis on $PM_{2.5}$ TFV and TFM, resulting
in the spatial structure of $PM_{2.5}$ transport flux under the temporal disturbances at the synoptic scale,
and revealing the connection between synoptic-scale disturbances in the EAWM and QWO in
regional $PM_{2.5}$ transport in CEC. Our study focuses on the driving effects of synoptic-scale
disturbances associated with cold air activity with the anomalous northerly winds in EAWM on
QWO of regional $PM_{2.5}$ transport over CEC, exacerbating $PM_{2.5}$ pollution in the downwind THB.

Differently from the studies on stagnant meteorological conditions associated with $PM_{2.5}$ accumulations (Gao et al., 2020; Wu et al., 2023; Yang et al., 2024b), this study provides new insights into the understanding of regional $PM_{2.5}$ transport with source-receptor relationship with the meteorological mechanism in atmospheric environment change.

Based on the 5-winter (2015-2019) observations of $PM_{2.5}$ concentrations and the corresponding meteorological reanalysis data, this study with the climate statistical and diagnostic methods investigates the QWO of regional $PM_{2.5}$ transport in China with the influence of synoptic-scale disturbance of EAWM circulation, providing a new insight into the understanding of regional air pollutant transport with meteorological drivers in atmospheric environment changes. Besides the EEOF method used in this study, the alternative methods of wavelet analysis, power spectrum analysis, and band-pass filtering could be used in further study. Future studies with utilizing long-term observations of air pollutants and meteorology over CEC could more comprehensively understand the variations in the regional transport of particles and the gaseous precursors with their contributions to air pollution, through the integration of artificial intelligence and physical-chemical process analyses.

*Data availability.* All data used in this paper can be provided upon request from Yongqing Bai (2007byq@163.com)

*Author contributions.* YB and TZ conceived the study. YB designed the graphics and wrote the manuscript with help from TZ, KM, YZ, JX, XS, LS, YY, YZ, WH and JY were involved in the scientific discussion. All authors commented on the paper.

*Competing interests.* The authors declare that they have no conflict of interest.

*Financial support.* This research was supported by the National Natural Science Foundation of China (grant no. 42075186, 41830965) and the National Key Research and Development Program of China (2022YFC3701204).

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
