# Peer review of "Quasi-weekly oscillation of regional PM2.5 transport over 1 China driven by the synoptic-scale disturbance of East 2 Asian Winter Monsoon circulation 3 Yongqing Bai 1, Tianliang Zhao 2,\*, Kai Meng 3,\*, Yue Zhou 1, Jie Xiong 1</sup"

_EGUsphere, 2024_

## Author Comment (AC1)

Dear Editors and Reviewers,

Thank you very much for your careful review and helpful comments on our manuscript egusphere-2024-2493. We appreciate very much your constructive comments and encouraging suggestions on our manuscript. We have accordingly made the careful revisions. The revised portions are highlighted in the revised manuscript. Please find our point to point responses to the reviewer's comments as follows:

**Responses to the reviewer 2**

*The regional transport of air pollutants driven by atmospheric circulation is one of the important causes for air pollution in atmospheric environment change. However, the driving mechanisms of atmospheric circulation have been poorly understood in the variations of regional Pair pollutant transport. Therefore, it is interesting in this study on the quasi-weekly oscillation (QWO) of regional $PM_{2.5}$ transport over China regulated with the influence of synoptic-scale disturbance of the monsoon circulation. The paper is well organized and fit the scope of the ACP journal. There are some minor issues need to be improved before publication.*

*[1. The quasi-weekly oscillation (QWO) of the monsoon circulation is a driver for the regional transport of air pollutants. Please include a discussion regarding the quasi-weekly oscillation of the East Asian winter monsoon circulation system.]*

**Response 1:** Many thanks for the encouraging comments and helpful suggestions on our manuscript. Following the reviewer's comments, we have accordingly revised the manuscript in lines 177-186.

Atmospheric motion encompasses a variety of temporal and spatial scales. The sequences of meteorological variables often contain complex periodic components and exhibit multi-time-scale variations, including daily, weekly, seasonal, and interannual variations. Numerous observations have found QWO with periods of less

than 10 days across various meteorological elements in the EAWM system (Compo et al., 1999; Murakami, 1979; Wu and Wang, 2002). Synoptic-scale atmospheric variations are closely related to atmospheric longwave adjustments, with QWO periods of 4-7 days observed in cold air activities of the EAWM (Bai et al., 2022; Wu and Wang, 2002). The synoptic-scale disturbance regulates the generation, transport, and removal of $PM_{2.5}$ in air pollution, which is a key mechanism behind the 4-7 day periodic changes in $PM_{2.5}$ in CEC during the periods of EAWM (Guo et al., 2014; Liu et al., 2018; Quan et al., 2014, 2020).

**Refrences**

Bai, Y., Zhao, T., Hu, W., Zhou, Y., Xiong, J., Wang, Y., Liu, L., Shen, L., Kong, S., Meng, K., and Zheng, H.: Meteorological mechanism of regional $PM_{2.5}$ transport building a receptor region for heavy air pollution over Central China. Sci. Total Environ., 808, 151951, https://doi.org/10.1016/j.scitotenv.2021.151951, 2022.

Compo, G. P., Kiladis, G. N. and Webster, P. J.: The horizontal and vertical structure of east Asian winter monsoon pressure surges. Quart. J. Roy. Meteor. Soc., 125, 29–54, https://doi.org/10.1256/smsqj.55302, 1999.

Guo, S., Hu, M., Zamora, M. L., Peng, J., and Zhang, R.: Elucidating severe urban haze formation in China. P. Natl. Acad. Sci. USA, 111, 17373–17378, https://doi.org/10.1073/pnas.1419604111, 2014.

Liu, Q., Jia, X., Quan, J., Li, J., Li, X., Wu, Y., Chen, D., Wang, Z., and Liu, Y.: New positive feedback mechanism between boundary layer meteorology and secondary aerosol formation during severe haze events. Sci. Rep., 8, 6095, https://doi.org/10.1038/s41598-018-24366-3, 2018.

Murakami, T.: Winter monsoonal surges over East and Southeast Asia. J. Meteor. Soc. Japan., 57, 133–158, https://doi.org/10.2151/jmsj1965.57.2_133, 1979.

Quan, J., Tie, X., Zhang, Q., Liu, Q., Li, X., Gao,Y., and Zhao, D.: Characteristics of heavy aerosol pollution during the 2012–2013 winter in Beijing, China. Atmos. Environ., 88, 83–89. https://doi.org/10.1016/j.atmosenv.2014.01.058, 2014.

Quan, J., Xu, X., Jia, X.. Liu, S., Miao, S., Xin, J., Hu, F., Wang, Z., Fan, S., Zhang, H., Mu, Y., Dou, Y., and Cheng, Z.: Multi-scale processes in severe haze events in China and their interactions with aerosols: Mechanisms and progresses. Chin. Sci. Bull., 65, 810–824, https://doi.org/10.1360/tb-2019-0197, 2020.

Wu, B., and Wang, J.: Winter Arctic oscillation, Siberian High and EAWM, Geophys. Res. Lett., 29, 1897, https://doi.org/10.1029/2002gl015373, 2002.

*[2. It would strengthen the manuscript to include a quantitative comparison of the changes in PM₂.₅, specifying the extent of any reductions or increases. A clear data comparison can make the manuscript more rigorous.]*

**Response 2:** Following the reviewer's comments, we have added the following content (lines 317-323) and Figure S2:

Under the context of QWO, the average $PM_{2.5}$ TFM in NCP decreases from approximately 400 μg m$^{-2}$ s$^{-1}$ in the 1st and 2nd phases to 200 and 100 μg m$^{-2}$ s$^{-1}$ in the 3rd and 4th phases, respectively (Fig. S2a). Correspondingly, the $PM_{2.5}$ concentration anomalies decline from around 100 μg m$^{-3}$ to approximately –50 μg m$^{-3}$ (Fig. S2c). In the downwind THB, the average $PM_{2.5}$ TFM increases from about 200 μg m$^{-2}$ s$^{-1}$ in the 1st phase to approximately 300 μg m$^{-2}$ s$^{-1}$ in the 2nd and 3rd phases (Fig. S2b), with $PM_{2.5}$ concentration anomalies also rising to around 50 μg m$^{-3}$ (Fig. S2d).

[Figure]

Figure S2. Box plots illustrating the 8 phases of QWO during 23 typical events of regional PM$_{2.5}$ transport from NCP to THB of (a,b) PM$_{2.5}$ TFM (µg m$^{-2}$ s$^{-1}$) and (c,d) anomalies of PM$_{2.5}$ (unit: µg m$^{-3}$); each box plot displays the maximum, minimum, median, and upper and lower quartiles, with the circles indicating the mean.

*[3. The manuscript provides a detailed analysis of regional PM$_{2.5}$ transport from the North China Plain to the Twain-Hu Basin; however, there is insufficient discussion regarding the impact of this transport on PM$_{2.5}$ over the Twain-Hu Basin. Please include a paragraph in the discussion that addresses potential impacts and offers a comparison with existing literature on this topic. ]*

**Response 3:** Following the reviewer's comments, we have accordingly revised the manuscript in lines 381-391.

Driven by prevailing winds of EAWM, the THB became the main receptor for regional transport of air pollutants over CEC (Bai et al., 2022; Shen et al., 2021). During 2015–2019, approximately 65.2% of the total PM$_{2.5}$ heavy pollution events in the THB were triggered by regional transport of air pollutants over CEC (Hu et al., 2022; Shen et al., 2021). Such PM$_{2.5}$ transport from upstream source regions in CEC contributes 51%-85.7% of the PM$_{2.5}$ pollution over the THB receptor region (Hu et al., 2021; Lu et al., 2017; Shen et al., 2022; Yu et al., 2020), revealing the dominance of regional transport of air pollutants from CEC to the THB with the meteorological drivers. Our research emphasizes the QWO of regional PM$_{2.5}$ transport over CEC with the driver of the synoptic-scale disturbances of EAWM circulation, confirming the source-receptor relationships with their 2-day lagging effects in the regional PM$_{2.5}$ transport between the upstream NCP source region and the THB receptor region.

**References**

Bai, Y., Zhao, T., Hu, W., Zhou, Y., Xiong, J., Wang, Y., Liu, L., Shen, L., Kong, S., Meng, K., and Zheng, H.: Meteorological mechanism of regional $PM_{2.5}$ transport building a receptor region for heavy air pollution over Central China. Sci. Total Environ., 808, 151951, https://doi.org/10.1016/j.scitotenv.2021.151951, 2022.

Hu, W., Zhao, T., Bai, Y., Kong, S., Xiong, J., Sun, X., Yang, Q., Gu, Y., and Lu, H.: Importance of regional $PM_{2.5}$ transport and precipitation washout in heavy air pollution in the Twain-Hu Basin over Central China: Observational analysis and WRF-Chem simulation. Sci. Total Environ., 758, 143710, https://doi.org/10.1016/j.scitotenv.2020.143710, 2021.

Hu, W., Zhao, T., Bai, Y., Kong, S., Shen, L., Xiong, J., Zhou, Y., Gu, Y., Shi, J., Zheng, H., Sun, X., and Meng, K.: Regulation of synoptic circulation in regional $PM_{2.5}$ transport for heavy air pollution: Study of 5-year observation over central China. J. Geophys. Res.-Atmos., 127, e2021JD035937, https://doi.org/10.1029/2021JD035937, 2022.

Lu, M., Tang, X., Wang, Z., Gbaguidi, A., Liang, S., Hu, K., Wu, L., Wu, H., Huang, Z., and Shen, L.: Source tagging modeling study of heavy haze episodes under complex regional transport processes over Wuhan megacity, Central China. Environ. Pollut., 231, 612–621, https://doi.org/10.1016/j.envpol.2017.08.046, 2017.

Shen, L., Hu, W., Zhao, T., Bai, Y., Wang, H., Kong, S., and Zhu, Y.: Changes in the Distribution Pattern of $PM_{2.5}$ Pollution over Central China. Remote. Sens., 13, 4855, https://doi.org/10.3390/rs13234855, 2021.

Shen, L., Zhao, T., Liu, J., Wang, H., Bai, Y., Kong, S., and Shu, Z.: Regional transport patterns for heavy $PM_{2.5}$ pollution driven by strong cold airflows in Twain-Hu Basin, Central China. Atmos. Environ., 269, 118847, https://doi.org/10.1016/j.atmosenv.2021.118847, 2022.

Yu, C., Zhao, T., Bai, Y., Zhang, L., Kong, S., Yu, X., He, J., Cui, C., Yang, J., You, Y., Ma, G., Wu, M., and Chang, J.: Heavy air pollution with a unique "non-stagnant" atmospheric boundary layer in the Yangtze River middle basin aggravated by regional transport of $PM_{2.5}$ over China. Atmos. Chem. Phys., 20, 7217–7230, https://doi.org/10.5194/acp-20-7217-2020, 2020.

*Minor comments:*

*[4. Line 45: The citation "X. Huang et al., 2020" appears to be formatted incorrectly. A similar issue is observed in lines 88-91 with "Y. Yang et al., 2021" and "W. Yang et al., 2021." Please correct these references and conduct a thorough check of the entire manuscript to ensure the accuracy of the citation format. ]*

**Response 4:** Thanks the referee for pointing out the printing errors, which have been corrected in the revised manuscript.

*[5. Line 93: The authors should provide a more detailed explanation of the term "susceptibility zone" to enhance comprehension of this concept. ]*

**Response 5:** Following the reviewer's comments, we have provided a more detailed explanation of the term "susceptibility zone" in lines 102-106.

The "harbor" effect on the eastern lee of the Tibetan Plateau's large topography on the westerlies is possibly an important factor influencing the regional distribution of $PM_{2.5}$ pollution in CEC with weak horizontal winds and sinking motion in the lower troposphere, which exacerbates the environmental impacts of local air pollutant emissions establishing a "susceptibility zone" in this region.

*[6. Lines 289-290: The sentence "The source-receptor relationship between NCP and THB during the regional $PM_{2.5}$ transport over CEC is discussed in detail in the next section" does not significantly contribute to the manuscript. As it serves primarily as a transition to the next section, it could be removed for clarity. ]*

**Response 6:** Following the reviewer's comments, we have removed the sentence from the revised manuscript.

*[7. Please improve English writing include grammar and expression in the manuscript. ]*

**Response 7:** With the help of English Language editing service, the English witting errors including incorrect grammar, confusing wording and inappropriate expression have been substantially revised to improve the readability of the manuscript.

---

## Author Comment (AC2)

Dear Editors and Reviewers,

Thank you very much for your careful review and helpful comments for improving our manuscript egusphere-2024-2493. We have accordingly made the careful revisions. The revised portions are highlighted in the revised manuscript. Please find our point to point responses to the reviewer's comments as follows:

**Responses to the reviewer 3**

*[This study investigates how the synoptic-scale circulations influence the transport of air pollutants from North China Plain to central and eastern China. They found the quasi-weekly oscillation of the air pollutants in North and East China is mainly attributed to both East Asia winter monsoon disturbance and the periodic activities of Siberian High. I enjoyed reading this manuscript and found its topic is very interesting. The authors gave a clear description of their methods and results. I have only minor comments, as shown below:]*

**Response:** We appreciate very much your constructive comments and encouraging suggestions on our manuscript. We have accordingly made the careful revisions. The revised portions are highlighted in the revised manuscript. Please find our point to point responses to the reviewer's comments as follows:

*[1. Butterworth filter: I do not understand this technique. I suggest the authors could add more explanation of this method.]*

**Response 1:** Thank you for your suggestions. We have refined the explanation of atmospheric periodic variations and the Butterworth filter method in Section 2.3 (lines 177-202):

2.3 Butterworth filter

   Atmospheric motion encompasses a variety of temporal and spatial scales. The

sequences of meteorological variables often contain complex periodic components and exhibit multi-time-scale variations, including daily, weekly, seasonal, and interannual variations. Numerous observations have found QWO with periods of less than 10 days across various meteorological elements in the EAWM system (Compo et al., 1999; Murakami, 1979; Wu and Wang, 2002). Synoptic-scale atmospheric variations are closely related to atmospheric longwave adjustments, with QWO periods of 4-7 days observed in cold air activities of the EAWM (Bai et al., 2022; Wu and Wang, 2002). The synoptic-scale disturbance regulates the generation, transport, and removal of $PM_{2.5}$ in air pollution, which is a key mechanism behind the 4-7 day periodic changes in $PM_{2.5}$ in CEC during the periods of EAWM (Guo et al., 2014; Liu et al., 2018; Quan et al., 2014, 2020). Based on the research objectives, identifying the desired periodic components from the original observational sequences is referred to as sequence filtering. In this study, we employed a Butterworth filter to extract QWO from observational data.

The Butterworth filter is commonly used to separate atmospheric periodic variations across specific frequency bands. Due to its smooth amplitude response, linear phase characteristics, and ease of implementation, Butterworth filter has been widely applied in climate and meteorological studies (Gouirand et al., 2012; Yang et al., 2024a). The Butterworth filter can be configured as a low-pass, high-pass, or band-pass filter, depending on the specific requirements. A band-pass filtering only allows signals within a defined frequency range to pass through with attenuating signals outside the defined frequency range. It is often employed to extract and analyze signals within specific frequency bands, such as particular weather patterns and climate cycles. In this study, to investigate the QWO (8-d) of regional $PM_{2.5}$ transport over the CEC under the influence of EAWM circulations in the synoptic scale, we applied Butterworth band-pass filtering to the daily TFM of $PM_{2.5}$ change and daily SLP anomalies during the winters of 2015-2019 for identifying at the quasi-weekly (6-9 days) synoptic-scale component of regional transport of $PM_{2.5}$ over CEC.

**References:**

Bai, Y., Zhao, T., Hu, W., Zhou, Y., Xiong, J., Wang, Y., Liu, L., Shen, L., Kong, S., Meng, K., and Zheng, H.: Meteorological mechanism of regional $PM_{2.5}$ transport building a receptor region for heavy air pollution over Central China. Sci. Total Environ., 808, 151951, https://doi.org/10.1016/j.scitotenv.2021.151951, 2022.

Compo, G. P., Kiladis, G. N. and Webster, P. J.: The horizontal and vertical structure of east Asian winter monsoon pressure surges. Quart. J. Roy. Meteor. Soc., 125, 29–54, https://doi.org/10.1256/smsqj.55302, 1999.

Gouirand, I., Jury, M. R., and Sing, B.: An analysis of low- and high-frequency summer climate variability around the Caribbean Antilles. J. Climate, 25, 3942–3952, https://doi.org/10.1175/jcli-d-11-00269.1, 2012.

Guo, S., Hu, M., Zamora, M. L., Peng, J., and Zhang, R.: Elucidating severe urban haze formation in China. P. Natl. Acad. Sci. USA, 111, 17373–17378, https://doi.org/10.1073/pnas.1419604111, 2014.

Liu, Q., Jia, X., Quan, J., Li, J., Li, X., Wu, Y., Chen, D., Wang, Z., and Liu, Y.: New positive feedback mechanism between boundary layer meteorology and secondary aerosol formation during severe haze events. Sci. Rep., 8, 6095, https://doi.org/10.1038/s41598-018-24366-3, 2018.

Murakami, T.: Winter monsoonal surges over East and Southeast Asia. J. Meteor. Soc. Japan., 57, 133–158, https://doi.org/10.2151/jmsj1965.57.2_133, 1979.

Quan, J., Tie, X., Zhang, Q., Liu, Q., Li, X., Gao,Y., and Zhao, D.: Characteristics of heavy aerosol pollution during the 2012–2013 winter in Beijing, China. Atmos. Environ., 88, 83–89. https://doi.org/10.1016/j.atmosenv.2014.01.058, 2014.

Quan, J., Xu, X., Jia, X.. Liu, S., Miao, S., Xin, J., Hu, F., Wang, Z., Fan, S., Zhang, H., Mu, Y., Dou, Y., and Cheng, Z.: Multi-scale processes in severe haze events in China and their interactions with aerosols: Mechanisms and progresses. Chin. Sci. Bull., 65, 810–824, https://doi.org/10.1360/tb-2019-0197, 2020.

Wu, B., and Wang, J.: Winter Arctic oscillation, Siberian High and EAWM, Geophys. Res. Lett., 29, 1897, https://doi.org/10.1029/2002gl015373, 2002.

Yang, Q., Zhao, T., Bai, Y., Wei, J., Sun, X., Tian, Z., Hu, J., Ma, X., Luo, Y., Fu, W., and Yang, K.: Interannual variations in ozone pollution with a dipole structure over Eastern China associated with springtime thermal forcing over the Tibetan Plateau. Sci. Total Environ., 923, 171527, https://doi.org/10.1016/j.scitotenv.2024.171527, 2024a.

*[2. In equation (6): Is the distance d in the unit of meter/kilometer or degree in longitude and latitude cause the authors mentioned 0.25 by 0.25 grid spacing in Line 159.]*

**Response 2:** The term refers to degrees in longitude and latitude. We have revised it as follows (lines 171-172):

It is necessary to interpolate the station data of zonal and meridional components ($F_u$, $F_v$) of PM$_{2.5}$ TFV to grid spacing with **0.25 by 0.25 degree in longitude and latitude** in CEC and then calculate the divergence of PM$_{2.5}$ TF at each grid point according to Formula (6).

*[3. Line 204: Delete "found" ]*

**Response 3:** Thank you for pointing this out. *"found"* has been removed.

*[4. Line 69: It should be "have not yet" ]*

**Response 4:** Thank you for pointing this out. It has been corrected in the revised manuscript in line 69.

*[5. Line 71: Change it to "one of the most active" ]*

**Response 5:** Thank you for your suggestion. It has been corrected in the revised manuscript in line 71.

*[6. Lines 77–78: It is not clear. Do you mean the pollutants can be removed by circulations at the regional scale? ]*

**Response 6:** We apologize for the unclear expression in the previous version. It has been revised as follows (lines 77-80):

The rapid southward advance of cold air with strong Siberian High can effectively drive the regional transport of air pollutants with less accumulations across CEC, while the weak Siberian High with the slow southward movement of cold air can particularly favorable for the transport of air pollutants from the northern source regions to southern receptor region over CEC.

*[7. Line 101: Delete "be beneficial to". ]*

**Response 7:** Thank you for pointing this out. The *"be beneficial to"* has been removed.

*[8. Line 121–122: What observations 4 times a day are used? ERA5-land is not observational dataset. ]*

**Response 8:** Thank you for pointing out the error. The statement has been revised with deleting the "observations" as follows (lines 134-135):

The U- and V-components of the 10-m wind over CEC were **obtained at 00, 06, 12, and 18 UTC daily** during the winter (December-February) of 2015-2019.

*[9. Line 132: The units for $PM_{2.5}$ mass concentration TF is not right. Units in the current form is for mass TF.]*

**Response 9:** Thank you for pointing out the error. It should refer to $PM_{2.5}$ mass, not $PM_{2.5}$ mass concentration. The revision is as follows (line 145):

The horizontal $PM_{2.5}$ TF is defined as the $PM_{2.5}$ mass passing through a unit area per unit time (unit: $\mu g\, m^{-2}\, s^{-1}$).

*[10. Line 179: should be observations]*

**Response 10:** Thank you for your careful review. It has been corrected with "observations" in the revised manuscript in line 210.

*[11. Line 201–204: It needs to be revised. ]*

**Response11:** We apologize for the unclear expression. It has been revised as follows (lines 232-234):

EEOF is an extension of the EOF to analyze the autocorrelations of the variable field over time. By selecting a lag time, the original observational matrix is expanded into multiple continuous time matrices, diagnosing the temporal changes in the spatial structure of variable fields.

*[12. Figure 1: What do grid cells of white color mean in panel (b)? ]*

**Response 12:** The grid cells in white color represent "missing values." Due to the sparse distribution of observational stations in regions with complex terrain, interpolation to grid cells can cause the "distorted" values. To avoid affecting the analysis results, these "distorted" values were replaced with missing values, as described in the figure caption.

Figure 1. Spatial pattern of the (a) EOF1 and (b) EOF2 loadings of the daily changes in $PM_{2.5}$ TFV anomalies (vectors, unitless) and TFM anomalies (color contours, unitless) over CEC during the winters of 2015-2019. The red and blue boxes indicate the NCP and THB, respectively. **The grid cells in white represent "missing values".**

*[13. Line 245: Do you mean "temporal structure"? ]*

**Response 13:** We apologize for the unclear expression. This refers to "the temporal changes in the moving spatial structure of PM$_{2.5}$ TF", which has been corrected in the revised manuscript in line 281.

*[14. For figures showing shadings: please add units for all color bars in the figure.]*

**Response 14:** Thank you for your suggestions. Figures 1, 2, and S1 show spatial pattern of the EOF and EEOF dimensionless loads (unitless). Figures 3, 4, 6, 7, as well as figures S4 and S5, have been updated to include the units of PM$_{2.5}$ (unit: μg m$^{-3}$),PM$_{2.5}$ TFM (unit: μg m$^{-2}$ s$^{-1}$), Divergence of PM$_{2.5}$ flux (unit: $10^{-3}$ μg m$^{-3}$ s$^{-1}$) , Temperature (unit:℃) and SLP$_{QWO}$ (unit: hPa) on the color bars.

---

## Author Comment (AC3)

Dear Editors and Reviewers,

Thank you very much for your careful review and helpful comments on our manuscript egusphere-2024-2493. We appreciate very much your constructive comments and encouraging suggestions on our manuscript. We have accordingly made the careful revisions. The revised portions are highlighted in the revised manuscript. Please find our point to point responses to the reviewer's comments as follows:

**Responses to the reviewer 1**

*[This paper investigates the quasi-weekly oscillation of $PM_{2.5}$ transport over China. The period analysed extended during the winters of 2015-2019 and $PM_{2.5}$ concentrations included data from 1079 stations. The statistical procedure considered the extended empirical orthogonal function. Two transport flux patterns were identified and 8 phases of the regional $PM_{2.5}$ transport over central and eastern China were isolated. Moreover, the synoptic pattern influence on the quasi-weekly oscillation of the regional $PM_{2.5}$ transport was analysed. Since this is a complex study, the paper may be accepted for publication in Atmospheric Chemistry and Physics after the inclusion of the following minor changes.]*

**Response 1:** We appreciate very much your constructive comments and encouraging suggestions on our manuscript. We have accordingly made the careful revisions. The revised portions are highlighted in the revised manuscript. Please find our point to point responses to the reviewer's comments as follows:

*[The authors have identified this quasi-weekly oscillation with a complex statistical procedure. Potential readers could ask if this oscillation could be observed with simpler procedures. If this oscillation responds to an atmospheric pattern, perhaps it could be perceived by direct measurements. If this oscillation responds to an atmospheric pattern, the authors should highlight the main advantages of the employed procedure (extended empirical orthogonal function), which is not easy,*

*against other procedures. Moreover, the authors should indicate if alternative and simple procedures exist to get similar results. ]*

**Response 2:** Many thanks for the encouraging comments and helpful suggestions on our manuscript. Following the reviewer's comments, we have accordingly revised the manuscript in lines 232-234, lines 248-252 and lines 468-480.

[revised manuscript text omitted]

*[The authors should inform about processes similar to that described in this paper around the world. The number of references for discussion should be increased. They should be used to compare results of this study with those from previous analyses. Strengths and weaknesses of the current study should be included.]*

**Response 3:** Following the reviewer's comments, we have accordingly added the information about processes similar to that described in this paper (in Section 4, lines 468-480, lines 492-497), the number of references for discussions and the comparisons of results of this study with those from previous analyses (in Section 3.2, lines 381-391; in Section 4, lines 468-480), as well as the strengths and weaknesses of the current study (in Section 1, lines 92-101; in Section 4, lines 480-486) in the revised manuscript as follows:

[revised manuscript text omitted]

*[Minor remarks: L. 279. "H. Huang" should be "Huang". ]*

**Response 4:** Thanks the referee for pointing out the printing errors, which have been corrected in the revised manuscript.